# Nanopore Sequencing-Driven Mapping of Antimicrobial Resistance Genes in Selected *Escherichia coli* Isolates from Pigs and Poultry Layers in Nigeria

**DOI:** 10.3390/antibiotics14080827

**Published:** 2025-08-14

**Authors:** Akinlabi Oladele Ogunleye, Prakash Ghosh, Adja Bousso Gueye, Foluke Olajumoke Jemilehin, Adelekan Oluseyi Okunlade, Veronica Olatimbo Ogunleye, Rea Maja Kobialka, Finja Rausch, Franziska Tanneberger, Adebowale Titilayo Philip Ajuwape, Ousmane Sow, George Olusegun Ademowo, Ulrike Binsker, Ahmed Abd El Wahed, Uwe Truyen, Yakhya Dieye, Cheikh Fall

**Affiliations:** 1Department of Veterinary Microbiology, Faculty of Veterinary Medicine, University of Ibadan, Ibadan 200001, Nigeriafoljem@yahoo.com (F.O.J.); dr_seyi@yahoo.com (A.O.O.); adebotiti.ajuwape@gmail.com (A.T.P.A.); 2Institute for Animal Hygiene and Veterinary Public Health, Leipzig University, 04103 Leipzig, Germany; rea_maja.kobialka@vetmed.uni-leipzig.de (R.M.K.); finja.rausch@web.de (F.R.); ahmed.abd_el_wahed@uni-leipzig.de (A.A.E.W.); truyen@vmf.uni-leipzig.de (U.T.); 3Department of Empirical Health Economics, Technische Universität Berlin, 10623 Berlin, Germany; 4International Centre for Diarrhoeal Disease Research, (icddrb,b), Dhaka 1212, Bangladesh; 5Pole de Microbiologie, Institut de Pasteur de Dakar, 36 Av. Pasteur, Dakar 220, Senegal; adjabousso.gueye@pasteur.sn (A.B.G.); ousmane.sow@pasteur.sn (O.S.); yakhya.dieye@pasteur.sn (Y.D.); cheikh.fall@pasteur.sn (C.F.); 6Department of Medical Microbiology, University College Hospital, University of Ibadan, Ibadan 200001, Nigeria; lolaore@yahoo.com; 7Institute of Advanced Medical Research and Training, University College Hospital, University of Ibadan, Ibadan 200001, Nigeria; ademowo_g@yahoo.com; 8Department Biological Safety, German Federal Institute for Risk Assessment, 10589 Berlin, Germany; ulrike.binsker@bfr.bund.de

**Keywords:** nanopore sequencing, resistance genes, AMR surveillance, EPI2ME, *Escherichia coli*, animal

## Abstract

**Background:** Despite the huge burden of deaths associated with or attributable to antimicrobial resistance, studies on sequencing based antimicrobial resistance (AMR) monitoring in Africa are scarce, specifically in the animal sector. Objective and Methods: With a view to deploy rapid AMR monitoring through leveraging advanced technologies, in the current study, nanopore sequencing was performed with 10 *E. coli* strains isolated from rectal swabs of pigs and poultry layers in Nigeria. Two sequence analysis methods including command line, where bacterial genomes were assembled, and subsequently antimicrobial resistance genes (ARGs) were detected through online databases, and EPI2ME, an integrated cloud-based data analysis platform with MinION, was used to detect ARGs. **Results:** A total of 95 ARGs were identified and most of the genes are known to be expressed in the chromosome. Interestingly, few genes including *qnrS1*, *qnrS15*, *qnrS10*, *kdpE*, *cmlA1*, *MIR-14*, *sul3* and *dfrA12* were identified which were previously reported as transferred through Mobile Genetic Elements (MGEs). The antibiotic susceptibility assay determined that the *E. coli* isolates were resistant to Penicillin (100%), Ciprofloxacin (70%), tetracycline (50%) and Ampicillin (40%). The accuracies of the command line and EPI2ME methods have been found to be 57.14% and 32.14%, respectively, in predicting AMR. Moreover, the analysis methods showed 62.5% agreement in predicting AMR for the *E. coli* isolates. Conclusions: Considering the multiple advantages of nanopore sequencing, the application of this rapid and field-feasible sequencing technique holds promise for rapid AMR monitoring in low- and middle-income countries (LMICs), including Nigeria. However, the development of a robust sequence analysis pipeline and the optimization of the existing analysis tools are crucial to streamline the deployment of nanopore sequencing in LMICs for AMR monitoring both in animal and human sectors.

## 1. Introduction

Antimicrobial resistance (AMR) is a global public health problem that impacts all countries (developed or underdeveloped) and all people, regardless of their wealth or status. The magnitude of the AMR threat, and the need to contain and control it, is widely acknowledged by countries’ governments, international agencies, researchers and private companies alike [1]. The possibility of the circulation of drug-resistant bacteria or AMR genes across the different sectors, through contaminated food (along the food chain), the environment, or through direct contact are well documented [2,3]. Thus, the impact of the threat posed by AMR is not only limited to humans, but also involves animals and the environment, due to the emergence, spread, and persistence of multidrug-resistant (MDR) bacteria, which are commonly shared amidst the three sectors [4].

The magnitude of the antimicrobial resistance (AMR) problem among bacterial pathogens is often not well captured in some parts of Africa, including Nigeria [5,6]. For instance, a finding from the regional Fleming Fund initiative from 205 laboratories in 14 African countries that reviewed 819,584 AMR records revealed that few of them (1.3%) undertook bacteriological testing routinely. Moreover, 80% of them had conducted less than 1000 antimicrobial susceptibility tests, and only a third of the WHO priority bacterial pathogen list was consistently tested in these countries [6].

The limited information is attributable to the fact that the surveillance of drug resistance is carried out in only a few countries, resulting in a scarcity of accurate and reliable data on AMR in general, and on antibiotic resistance in particular. This applies to many common and serious infectious diseases that are prevalent across the continent, such as meningitis, pneumonia and bloodstream infections [7]. A more recent report on AMR in the WHO Africa region in 2019 reported an estimated 1.05 million deaths (95% UI 82,900–1,316,000) associated with bacterial AMR and 250,000 deaths (192,000–325,000) attributed to bacterial AMR. The study attributed the largest fatal AMR burden to lower respiratory and thorax infection (119,000 deaths (92,000–151,000) [8]. *Escherichia coli* was among the seven leading pathogens that were collectively responsible for 821,000 deaths (636,000–1,051,000) associated with resistance in the region [8].

AMR has been identified as one of the prominent factors that contributes towards compromising the effective management of a number of diseases in sub-Saharan Africa, including diseases spread by the fecal–oral route and other diarrheal diseases [9]. The enterotoxigenic, enteropathogenic and enteroaggregative *Escherichia coli* types have been reported as aetiologic agents of diarrheal diseases in Gabon, Kenya, Nigeria, Senegal and Tanzania [10,11]. It was estimated that the death rate attributable to AMR in 2019 was highest in western sub-Saharan Africa, reaching a burden of 27.3–114.8 deaths per 100,000 population. According to the same authors, the deaths attributable to and associated with bacterial antimicrobial resistance in Africa are higher than that observed in other regions such as East Asia, western Europe and Central Latin America [6,12]. In 2019, there were 64,500 deaths attributable to AMR in Nigeria and 263,400 deaths associated with AMR. Nigeria was reported to have the 185th highest age-standardized mortality rate per 100,000 population associated with AMR across 204 countries (https://www.healthdata.org/, accessed on 3 May 2025). Food animals such as pigs and poultry have been incriminated as sources of the development of antimicrobial resistance and the transfer of drug resistant microorganisms between food-producing animals and humans [13,14]. Some sequence types of *E. coli* that have been reported in animals from Nigeria including ST 131, ST 155, ST 48, ST 10, ST 215, ST 46, as well as ST 218, signifying the genomic diversities among the *E. coli* from food animals in Nigeria [15]. The ESBL-producing *E. coli* strains isolated from poultry in Nigeria include *bla_CTX-M-15_*, *bla_CTX-M-27_*, *bla_CTX-M-55_* and *bla_CTX-M-65_* [15,16]. The monitoring of AMR patterns in pathogenic bacterial isolates is therefore critical for the optimization of effective treatment guidelines for pathogens.

Conventional culturing and sensitivity laboratory procedures for the detection of drug-resistant pathogens are still the gold-standard techniques to screen and monitor drug-resistant pathogens in Nigeria. This method is time consuming, requiring at least 4 days, and requires qualified/experienced staff in designated laboratories [17]. Molecular diagnostics, especially whole genome sequencing, can be a valuable addition to AMR surveillance and provide information on the early emergence and spread of AMR and further support policy strategies on AMR control. However, most next-generation sequencing platforms are resource demanding and time consuming. On the other hand, the advantage of real-time whole genome sequencing diagnostic methods such as Oxford Nanopore Technology (ONT) over conventional bacteria pathogen isolation and identification as well as antibiotic resistance profiling is their scientifically established speed [18,19].

A previous study reported the detection of AMR-encoding genes and plasmids within 10 min following bacterial whole genome sequencing through MinION [20]. The study also showed that the detection of AMR genes from samples is possible within 1 h following nanopore sequencing [20]. The aforementioned evidence supports the application of ONT over conventional cultural identification and sensitivity methods for the detection of antimicrobial resistance. Therefore, in this study we applied nanopore sequencing followed by comparative genomic analysis to elucidate the resistance patterns in 10 randomly selected *E. coli* isolates from healthy commercial poultry and pigs slaughtered for consumption in Nigeria.

## 2. Material and Methods

### 2.1. Ethical Approval

All animal procedures and experimental protocols were approved by the University of Ibadan Animal Care and Use Research Ethics Committee (UI-ACUREC). All procedures and experimental protocols were carried out in accordance with the guidelines approved by the review committee with the approval number: UI-ACUREC/13-1221/9. Consent was obtained from farm owners in Ibadan prior to sample collection. The ARRIVE guidelines 2.0 were followed in the reporting of this study.

### 2.2. Sample Collection

Sampling was carried out from healthy pigs and poultry between January 2018 and March 2020 in Ibadan, Oyo state, and some parts of Osun and Ondo states, Nigeria. Therefore, 384 rectal swabs were collected aseptically from pigs just before slaughter, and 500 cloacal swabs from laying poultry layers between 22 and 50 weeks of age.

Samples were labeled and kept cool in a sterile jar for their shipment as early as possible to the bacteriological laboratories at the University College Hospital/Department of Veterinary Microbiology, University of Ibadan, for their microbiological analysis. All samples were accompanied by questionnaire forms to capture relevant information. The sampling process and rest of the laboratory procedures are delineated in Figure 1.

### 2.3. Culture and Isolation of Enterobacteriaceae

The collected samples were processed within 24 h to reduce contamination risks and to ensure viability of targeted organisms. The rectal swabs from pigs and cloacal sampling from chicken were cultured in 5 mL buffered peptone water (BPW) and incubated at 37 °C for 18 to 24 h. Subsequently, the broth cultures were sub-cultured on XLD plates and the rest poured into 10 mL of Selenite F broth, before being incubated overnight at 37 °C. From the Selenite F broth, the growing organisms were sub-cultured on XLD and MacConkey agars and incubated for 18 h at 37 °C [21]. Isolates suspected to belong to the family *Enterobacteriaceae* were purified and characterized or stored at −80 °C in tryptone soy broth for further investigations.

### 2.4. Identification of Isolates

Suspected bacteria colonies from the family *Enterobacteriaceae* were purified prior to Gram staining and an oxidase test to rule out *Pseudomonas* species. Therefore, Gram-negative and oxidase-negative isolates were further screened for ID species determination using API 20E gallery tests (Biomerieux, Lyon, France). Interpretation was performed based on the API web scoring software (apiweb^TM^) [18,19,20].

### 2.5. Antibiotics Susceptibility Test (AST)

The antibiotic susceptibility testing was performed using Mueller Hinton (MH) agar according to the Kirby–Bauer method. Briefly, a discrete colony from the pure culture was collected and homogenized in sterile distilled water. The turbidity of the suspension was determined and adjusted to an optical density of 0.5 McFarland, prior to its inoculation onto MH agar plate using dry swabs. In this study, AST was applied on *E. coli* isolates and inoculation was performed in duplicate and plates were incubated overnight at 37 °C. The following antibiotic disks were used: CN = Gentamicin (10 μg); CIP = Ciprofloxacin (5 μg); CAZ = ceftazidime (30 μg); AMC = Amoxicillin–Clavulanate (30 μg); SXT = Trimethoprim–Sulfamethoxazole (25 μg); C = Chloramphenicol (30 μg); AMP = Ampicillin (10 μg); TE = tetracycline (30 μg); CTX = Cefoxatime (30 μg); TOB = Tobramycin (10 μg); AK = Amikacin (30 μg); ETP = Ertapenem (10 μg); KZ = Cephazolin (30 μg); FEP = Cefepime (30 μg); NOR = Norfloxacin (10 μg); FOS = Fosfomycin (50 μg); S = Streptomycin (25 μg); TIC = Ticarcillin (75 μg); ATM = Aztreonam; P = Penincillin (10 units); IPM = Imipenem (10 μg); DO = Doxycycline (30 μg); TZP = Piperacillin–Tozobactam (110 μg); FOX = Cefoxitin (30 μg) and CRO = ceftriaxone (30 μg). The diameter of inhibition surrounding the disks were measured and the results were reported as S (sensitive), and both isolates that exhibited intermediate resistance and those that showed resistance were reported as being resistant R (resistant), according to the CLSI guideline, version 2019 (CLSI 2016). The ESBL phenotype was confirmed by the double-disk synergy test (DDST), the disk of cefotazime and ceftazidime alone and in combination with the clavulanic acid disk. An enhanced inhibition zone diameter of ≥5 mm produced by the combination disk compared with the respective single disk alone was used to confirm the ESBL production phenotypically. Isolates resistant to at least three different classes of antibiotics were considered as multidrug resistant (MDR). For quality control, *Escherichia coli* ATCC 25922 and *Klebsiella pneumoniae* ATCC 700603 were used as reference strains.

### 2.6. Bacterial DNA Extraction

DNA samples were extracted using the Qiagen DNeasy Blood and Tissue kit, according to the manufacturer’s recommendations. Test organisms were streaked from the glycerol tryptone broth stock onto MacConkey agar so that single colonies of isolates were obtained. It was incubated at 35–37 °C for 16–18 h. The growth on MacConkey agar was sub-cultured onto Eosine Methylene Blue (EMB) agar and incubated at 37 °C for 16–18 h. Subsequently, 2–4 pure isolated colonies typical of *Escherichia coli* were picked and homogenized into the lysis buffer, followed by DNA purification and collection into tubes. A sample with sterile water was used as the blank control to monitor background DNA. The quality and concentration of extracted DNA were evaluated using the Qubit 3.0 fluorometer (ThermoFisher Scientific, Waltham, MA, USA) with the 1X dsDNA HS Assay kit (Invitrogen, Carlsbad, CA, USA) and adjusted when needed for library preparation.

### 2.7. Library Preparation and Sequencing

For library preparation, the SQK-RBK110.96 protocol for rapid barcoding (Oxford Nanopore Technologies, Oxford, UK) was used. Barcoding was performed by ligation by mixing 9 μL of each sample containing a minimum of 50 ng template DNA with 1 μL of Rapid Barcodes (RB01-96, Oxford Nanopore Technologies, Oxford, UK) and incubation at 30 °C for 2 min and 80 °C for 2 min. Thereafter, samples were pooled and mixed with an equal volume of SPRI beads (Oxford Nanopore Technologies, Oxford, UK), followed by several washing steps with 80% ethanol. The trapping of impurities was performed using a magnetic rack, prior to the addition of 15 μL of Elution Buffer (Oxford Nanopore Technologies, Oxford, UK). The Flow Cell was primed by the Flush Tether and Flush Buffer containing priming mix (Oxford Nanopore Technologies, Oxford, UK). Finally, the library was prepared by mixing 37.5 μL of Sequencing Buffer, 25.5 μL of Loading Beads (Oxford Nanopore Technologies, Oxford, UK) and 12 μL of DNA library, before being loaded into the MinION Flow Cell (Oxford Nanopore Technologies, Oxford, UK) for sequencing in a MinION device (Oxford Nanopore Technologies, Oxford, UK).

### 2.8. Data Processing with Command Line Tools

#### 2.8.1. Quality Control

Raw reads generated as the fast5 files format were basecalled and demultiplexed using Guppy v3.6.1 and then transformed into fastq format using an adjusted parameter of 1000 reads per file. Quality control was thereafter applied using pycoQC v2.5.0.3 software and reads were trimmed using Porechop. The acceptable Phred-quality score was ≥7 and read size was ≥1000 nucleotides. De novo assembly was performed with Flye V.2.9.1 without polishing to minimize reads loss during quality filtering. Since most of the reads were less than 1000 nucleotides, all were considered for analysis, regardless of their size.

#### 2.8.2. Species Identification

Species identification was performed directly from the raw reads by querying a sketched version of the NCBI RefSeq database, which had been compressed to reduce its size and speed up comparisons. The tool Mash v1.1 was used to estimate the genetic distance between the reads of interest and reference genomes in the database. The species associated with the lowest genetic distance (≤0.05–0.07) was considered to be the identity of the analyzed strain.

### 2.9. Genome Analysis for Antimicrobial Resistance Prediction

Bacterial genome assemblies were screened against numerous databases and tools for antimicrobial resistance (AMR) gene characterization. Therefore, a compilation of local datasets was created, integrating antimicrobial resistance genes from relevant AMR databases, including NCBI, ARG-ANNOT, AMRFinder and CARD using a complete resistance database catalog. Each tool has its strengths and limitations and our local dataset collection was refined by deleted duplicated genes. Finally, a collection of 12,154 ARGs was used for screening using blast v2.12.0. Setting parameters were defined as 60% for coverage and 90% for gene identity.

Similarly, a collection of 460 plasmid replicon types was used to check putative plasmids using blast v2.12.0. Settings parameters were defined as 60% for coverage and 90% for species identity. As most of the reads were less than 1000 nucleotides, the assembling with Flye v.2.9.1 did not allow reliable outputs.

### 2.10. Data Processing with the Cloud-Based Platform EPI2ME

Using a MinION Flow Cell (Oxford Nanopore Technologies, Oxford, UK), the sequencing run was accomplished after 48 h. The generated sequence files were saved as FASTQ files on a laptop.

Online analysis was carried out using the “What’s in my pot” (WIMP) software available through the EPI2ME desktop application developed by Oxford Nanopore Technologies, UK. For this, the reads were aligned with the genome database of the NCBI. WIMP also conducted quality control that included, for example, sequence length information. The second analysis identified genes responsible for antimicrobial resistance. A filter was used to highlight clinically relevant results.

## 3. Results

### 3.1. Bacteriological Results

The present study revealed the diversity of Enterobacterial species, which was very high among both chickens (317 isolated from 500 sampled) and pigs (405 from 384 sampled) (Figure 2).

Common bacteria species from poultry include *E. coli* (*n* = 189; 59.7%); *Proteus mirabilis* (*n* = 40; 12.7%); *Klebsiella pneumoniae* (*n* = 23; 7.4%); *Enterobacter roggenkan* (*n* = 16; 5.0%); *Enterobacter homaechae* (*n* = 1; 0.3%); *Salmonella arizonae* (*n* = 12; 3.8%); *Serratia* sp. (*n* = 8; 2.5%); *Providencia stuartii* (*n* = 7; 2.2%); *Raoultella ornithinolytica* (*n* = 6; 1.9%); *Citrobacter werkmanii* (*n* = 3; 0.9%); *Klebsiella oxytoca* (*n* = 3; 0.9%); *Escherichia vulneris* (*n* = 1; 0.3%); *Escherichia fergusoni* (*n* = 1; 0.3%); *Morganella morganii* (*n* = 2; 0.6%); *Pantoea* sp. (*n* = 2; 0.6%); *Cronobacter* sp. (*n* = 1; 0.3%); *Kluyvera* sp. (*n* = 1; 0.3%) and *Shigella* sp. (*n* = 1; 0.3%).

Whereas, the 405 isolated from the pigs sampled include *E. coli* (*n* = 27; 68.9%); *Proteus mirabilis* (*n* = 19; 4.7%); *Klebsiella pneumoniae* (*n* = 13; 3.2%); *Enterobacter roggenkan* (*n* = 25; 6.2%); *Enterobacter homaechae* (*n* = 3; 0.74%); *Salmonella arizonae* (*n* = 8; 1.97%); *Serratia* sp. (*n* = 36; 8.9%); *Providencia stuartii* (*n* = 7; 1.7%); *Raoultella ornithinolytica* (*n* = 3; 0.74%); *Citrobacter werkmanii* (*n* = 2; 0.49%); *Klebsiella oxytoca* (*n* = 2; 0.49%); *Pantoea* sp. (*n* = 6; 1.48%) and *Hafnia alve* (*n* = 2; 0.49%).

The antimicrobial susceptibility testing on 10 *E. coli* isolates revealed low or no resistance for critical antibiotics such as cephalosporins (≤20%), aminoglycosides (≤10%), chloramphenicol (20%), sulfonamide (30%), beta-lactams (10–20%), carbapenems (≤10%) and phosphonic acid (10%). Nonetheless, significant resistance was observed for other antibiotics, such as Ampicillin (40%), tetracycline (50%), Ciprofloxacin (70%) and Penicillin (100%). The only isolate which showed resistance to cefotaxime and ceftazidime was not confirmed as an ESBL producer by DDST (Table 1).

### 3.2. Sequencing of the Bacterial Isolates

The 10 *E. coli* isolates were sequenced using Oxford Nanopore Technologies (ONT) for comparative analysis. These included nine isolates from pigs and one isolate from chickens. The prepared libraries were then pooled and loaded into a Flow Cell for whole genome sequencing. Depending on the analytical process, the reads generated were estimated to be 793 and 735, when using command lines and the EPI2ME platform, respectively, and corresponding sizes to be 1039 and 1780 bp.

The comparative analysis using WIMP v3.0.1 via the ONT-developed EPI2ME platform and the command line process, in reference to a custom collection from the Refseq database, respectively, showed good concordance for the species identification and classification, except for two cases. The discrepancies include an assignation as *Escherichia coli* by the WIMP approach in one case, which was considered as *Enterobacter hormaechei*, in reference to the local Refseq database collection. Moreover, one *Escherichia coli* could not be confirmed using the command line analytical process. The average turnaround time for a full bioinformatic process was estimated to be 75 min through the WIMP approach and 100 min when using the command line process. These covered quality control, reads assembly, annotation, species identification as well as plasmid, virulence and resistance factor characterization.

The comparison of predicted AMR determinants revealed differences between the command line and WIMP analytical processes. Overall, 95 ARGs encoding for 12 antibiotic families were retrieved. All of them were retrieved when using the command line approach, in contrast to the WIMP software (n = 58). In addition, the command line could not identify specific mutations pertaining to AMR. Notably, 11 mutations conferring presumptive resistance to antibiotics predicted by WIMP were not confirmed by command line. These include a mutation on the *acrR* repressor associated with high antibiotic resistance through an efflux pump mechanism, but also those on the 16S rRNA, including the *rrsB*, *rrsC*, *rrsD* and *rrsH* genes which may confer resistance to aminoside antibiotic classes such as fosfomycin, streptomycin and spectinomycin.

In spite of the limited number of isolates, correlations were retrieved for AMR prediction, regardless of the approach used (Figure 3).

### 3.3. Distribution of Antimicrobial Resistance Genes

Both command line and EPI2ME were used to predict ARGs, which were stratified against 12 antibiotic families. A total of 95 resistance genes were identified through sequence analysis. Following detection, the ARGs were classified according to the CARD database for ARGs ontology (https://card.mcmaster.ca, accessed on 5 May 2025). Moreover, the mechanism involving the resistance of the identified genes being retrieved from the CARD database are depicted in Figure 3 and Appendix A. In addition, the ARGs are known to be carried through mobile genetic elements (MGEs) are explained according to the ARGs ontology of the CARD database as well.

### 3.4. Fluoroquinolone Resistance Genes

Among the ten isolates, Fluoroquinolone ARGs were identified in nine isolates through command lines, whereas EPI2ME detected Fluoroquinolone resistance genes in seven isolates. In total, 25 ARGs encoding for Fluoroquinolone resistance were identified, but 14 of them were found exclusively using the command line (Figure 3). The other nine Fluoroquinolone resistance genes were identified through both of the analysis methods. The notable ARGs identified through command line include *emrA*, *emrR*, *mdtH*, *mdtM*, *mdtE*, *acrE*, *acrS*, *mdtF* and CRP which have been detected in at least 30% of the isolates. Through both of the analysis methods, the *emrB*, *evgA*, *gadX*, *acrF* and *evgS* genes were detected in at least 20% of the isolates. Three Fluoroquinolone resistance genes (*qnrS1*, *qnrS10* and *qnrS15*) have been described to be plasmid-mediated and the command line approach has been found to be more efficient for their detection in comparison to EPI2ME.

### 3.5. Aminoglycoside Resistance Genes

Most of the isolates were found to harbor aminoglycoside resistance genes according to the command line process, which is different when referred to the EPI2ME platform where ARGs were found in three isolates (Figure 3). In this study, a total of eight ARGs specific to aminoglycoside were detected. The genes *acrD*, *cpxA*, *baeS*, *tolC* and *aadA2* were detected by both analytical methods and expressed in at least one isolate. The plasmid-mediated gene, *kdpE,* which confers resistance to aminoglycoside, was detected in two isolates.

### 3.6. Penem Resistance Genes

Antibiotic resistance genes specific to Penem (n = 15) were detected in more than 50% of the isolates (Figure 3). They were retrieved among eight and six isolates by command line and EPI2ME, respectively. Common ARGs retrieved include *evgA*, *gadX*, *H-NS*, *tolC*, *acrF* and *evgS* genes. *Escherichia_coli_ampC* beta-lactamase were detected in one isolate by command line.

### 3.7. Phenicol Resistance Genes

More than 50% of the isolates were found to express ARGs specific to Phenicol resistance through the command line (Figure 3). In contrast, only one isolate expresses ARG encoding resistance to Phenicol when screened by EPI2ME. The common ARGs include the *mdtM*, *acrS*, *emrD* and *tolC* genes and retrieved among four isolates. The *tolC* gene was confirmed by both analysis methods. Elsewhere, resistance in two isolates against Phenicol can be attributed to the *cmlA1* gene which has been detected through the command line. This plasmid- or transposon-mediated gene is responsible for antibiotic efflux and is inducible by a low concentration of chloramphenicol.

### 3.8. Cephalosporin Resistance Genes

A total of 10 ARGs responsible for cephalosporin resistance was detected through sequence analysis (Figure 3). The command line allowed the detection of ARGs encoding resistance to cephalosporin in seven isolates, while these ARGs were identified among only three isolates using EPI2ME. The most frequent ARGs were *acrE* and *acrS* and retrieved mostly through the command line, whereas *H-NS*, *tolC* and *acrF* were identified through EPI2ME.

### 3.9. Cephamycin Resistance Genes

The sequence analysis explored five ARGs specific to cephamycin resistance in more than 50% of the isolates (Figure 3). *acrE* and *acrS* were found to be the most prevalent ARGs being identified in at least four isolates through the command line. The ARGs such as *H-NS*, *tolC* and *acrF* were identified through both analysis methods.

### 3.10. Tetracycline Resistance Genes

Most of the isolates harbored tetracycline resistance genes. They were retrieved among eight and five isolates through the command line and EPI2ME pipelines, respectively. Globally, 13 ARGs were detected, including the *emrY*, *evgA*, *H-NS*, *tolC*, *EvgS* and *oqxB* genes retrieved by both analysis methods (Figure 3).

### 3.11. Phosphonic Acid Resistance Genes

In this study, the *mdtG* gene was retrieved to predict resistance to phosphonic acid among three isolates by command line and one by EPI2ME application (Figure 3).

### 3.12. Monobactam Resistance Genes

The ARGs for the monobactam antibiotics family include the *marA* and *MIR-14* genes retrieved among few isolates (n = 02) through the command line analysis and absent using EPI2ME (Figure 3). Among the two genes, *MIR-14* is eventually acquired through plasmids.

### 3.13. Carbapenem Resistance Genes

Few isolates were found to harbor ARGs specific to carbapenem resistance. The gene *marA* was identified through command lines, while *tolC* was identified using both analysis methods (Figure 3).

### 3.14. Sulfonamide Resistance Genes

The *sul3* gene, eventually plasmid-located, was identified among two isolates and associated with sulfonamide resistance prediction (Figure 3). It was confirmed by both methods. Elsewhere, *dfrA12*, an integron-mediated gene conferring the resistance against Trimethoprim–Sulfamethoxazole was detected as well in two other isolates using the command line approach. This was confirmed by the WIMP online application.

### 3.15. Beta-Lactamase Resistance Genes

In this study, four isolates expressed ARGs specific to beta-lactamase resistance with the bla_Ec15_ and bla_ACT58_ genes found in one isolate. The bla_Ec18_ gene was expressed in three isolates (Figure 3). All of these genes have been detected by the command line approach without any confirmation from EPI2ME analysis.

### 3.16. Resistance Genes Associated with Other Antimicrobials

Apart from the ARGs against the antibiotics used for the phenotype screening of the isolates, a significant number of ARGs against other antibiotic classes were detected. These included the *dfrA12* and *oqxB* genes, associated with diaminopyrimidine resistance antibiotics. The *oqxB* gene was also described as being associated with resistance to Nitrofurantoin. The *mdtM* gene, which can induce resistance to the Lincosamide antibiotic, was also detected. Additional ARGs include *Escherichia_coli_acrA*, *acrS*, *acrB*, *marA* and *tolC*, which may induce resistance to Rifamycin (Enterobacter_cloacae_*acrA*) and Glycylcycline (*Escherichia_coli_acrA*, *acrS*, *acrB*, *marA*, *tolC* and *oqxB*). The *evgA*, *evgS*, *gadX*, *mdtE*, *H-NS*, *mdtF*, *CRP* and *tolC* genes may be responsible for Macrolide resistance. The *mdtO*, *mdtP*, *mdtM* and *mdtN* genes are reported to be responsible for nucleoside antibiotic resistance, while *mdtB*, *cpxA*, *mdtA*, *baeS*, *tolC*, *mdtC* and *baeR* were described to be associated with aminocoumarin antibiotic resistance.

### 3.17. Correlation Between Phenotype and Genotype

The degree of correlation between the phenotype and genotype varied across the 12 antibiotic families. A comparison was conducted using 120 individual phenotypic results comprising 12 antibiotic families on 10 *E. coli* isolates. Based on the sequences analyzed, 67 ARGs were retrieved using the command line approach (G1) and 34 by EPI2ME (G2). Phenotypic resistance was observed in 28 cases (pink squares in Figure 4). AMR prediction was accurate in 16 cases (57.14%) with the command line approach and in 9 cases (32.14%) using EPI2ME. Taking together the sequence analysis approaches, the sensitivity to genotype was 71.42%, whereas the specificity was found to be 59.78%. A comparison between the two pipelines showed good concordance in 26 cases (62.5%), which dropped down to 6 cases (7.4%) when comparing genotyping to phenotypic results. The best match was for Fluoroquinolone (70%), unlike the beta-lactams, phosphonic acid and cephamycin antibiotic classes.

## 4. Discussion

To date, AMR data are poorly documented in Africa, due to inadequate and inefficient data collection and surveillance systems [22]. A survey from 14 African countries revealed that only 1.3% of laboratories undertook bacteriological testing and most of them are from the human health sector, with less than 1000 AST conducted annually [8]. Another survey revealed a high prevalence of antibiotic use among hospitals across sub-Saharan Africa, ranging from as low as 37.7% in South Africa to as high as 80.1% in Nigeria [23]. A similar situation was also found in the livestock sector from which antibiotic consumption from 13–27 countries was estimated to be 3558–4279 between 2015 and 2019. Most of them were for cattle and poultry production. In Nigeria, antibiotics were used at a rate of 36.9% among pig farmers and 20% among bird sellers for prophylaxis [9,24]. The overarching goal of this study was to map antimicrobial resistance genes among circulated bacteria from healthy poultry and pigs in Nigeria with a focus on *E. coli* isolates. Most of the selected isolates (60%) were resistant to at least three different classes of antibiotics, including tetracycline, cephalosporin, Fluoroquinolone, Penems, Phosphonic acid, sulfonamide and monobactam. These results are concordant with those from Tanzania, where isolated *E. coli* from poultry and domestic pigs showed a high resistance to tetracycline (63.5%), nalidixic acid (53.7%), Ampicillin (52.3%) and Trimethoprim/Sulfamethoxazole (50.9%). In Senegal, the *E. coli* colonization among healthy poultries and chicken carcasses between 2019 and 2021 was estimated to 68.2% and 68.3%, respectively, and the most common antibiotic resistance was represented by tetracycline (88.89–90.76%), sulfonamides (77.78–89.23%), Trimethoprim/Sulfamethoxazole complex (73.33–90.76%) and quinolones (70.46%) [25,26,27]. The circulation of multidrug-resistant (MDR) bacteria in Africa, in particular in the livestock sector, testifies an overuse of antibiotics for growth promoters with additionally little control on the quality of drug used. MDR could be mediated by mobile genetic elements (MGEs) such as transposons, integrons and/or plasmids [28]. In this study, generated reads did not allow the identification of MGEs with difficulties in correctly assembling plasmids. The average read size was between 1039 and 1780 pb and is below the capabilities of ONT, from which generated reads can exceed 1 megabase in length. Nonetheless, the mutation in the *acrR* gene, the repressor of the multidrug efflux complex AcrAB-TolC, could mediate the MDR to cefazolin, cefepime, ceftriaxone, tetracycline and aztreonam on one *E. coli* isolate. The mutation on the *acrR* gene is capable of inducing a high resistance to ciprofloxacin, tetracycline and ceftazidime, which could induce the MDR pattern on one *E. coli* type which is resistant to cefazolin, cefepime, ceftriaxone, tetracycline and aztreonam. Other mutations were mostly on the 16S rRNA on the *rrsB*, *rrsC*, *rrsD* and *rrsH* genes. Referring to the CARD repository (https://card.mcmaster.ca, accessed on 5 May 2025), a mutation on the above 16S rRNA gene could confer resistance to aminoglycosides, in particular to spectinomycin and streptomycin. In this study, only one *E. coli* type showed resistance to the tested aminoglycosides (amikacin and streptomycin). No mutation was retrieved from this isolate, but it harbored the aminoglycoside nucleotidyl transferase genes encoded by plasmids and/or integrons with the *aadA2* and *aadA15* genes. These genes were deciphered by both analytical processes. Other tested *E. coli* isolates, including MDR clones, harbored several other ARGs encoding resistance to different categories of antibiotics. The highest resistance rates were represented by ciprofloxacin (70%), tetracycline (50%) and cefazolin (40%), which can be mediated by efflux pump mechanisms such as *mdtEF-TolC* and *emrAB-TolC* complexes, which were frequently retrieved among the sequenced *E. coli*. They were described as being associated with multidrug resistance, including Fluoroquinolones, macrolides, penems and tetracyclines.

Surprisingly, no ESBL-producing *E. coli* were retrieved in this study; even so *blaEC-15* and *blaEC-18* were retrieved among three isolates using command line analysis. This contrasts with the latest trends in poultry farms and slaughterhouses across Africa and could be attributed to sampling bias. Studies from Tanzania, Senegal and Nigeria revealed a fecal carriage of ESBL-producing *E. coli* of 65.3% 60% 37.8%, respectively, among healthy poultry [25,29]. Even so, Erythromycin was not tested in this study, while a high prevalence of resistance to *E. coli* from poultry (MR 96.1%, IQR 83.3–100.0%) and pigs (MR 94.0%, IQR 86.2–94.0%) was recently documented across Africa [30].

It is urgent to change the paradigm of the tracking and management of AMR across Africa. Thus, the deployment of diagnostic precision tools, such as ONT, that can be used in the shortest time is valuable. Even so, the procurement of consumables and reagents remains a huge challenge, although whole genome sequencing has demonstrated its effectiveness in public health concerns. This was the case during the COVID-19 pandemic from which generated data, when combined with traditional epidemiological approaches, allowed early warning, identified unknown transmission chains and highlighted weakness in existing infection control measures [31]. Regional initiatives have been launched recently in Africa to bridge the gap on genomic surveillance, in particular on AMR. These include Africa-PGI and SeqAfrica [32].

Moreover, such information helps clinicians towards the prudent and judicious use of antibiotics. Like developed countries, sequence-based approaches are equally important for developing countries like Nigeria, where conventional microbiological tools are the mainstay to identify bacteria and for antibiotic susceptibility testing. Conventional culturing and sensitivity laboratory procedures for the detection of drug-resistant pathogens are still the usual means of screening and carrying out surveillance studies for the detection of drug-resistant pathogens in Nigeria. The method is time consuming as it usually requires 4–7 days or more depending on the pathogenic bacteria involved. The early detection of these drug-resistant pathogens is critical to control and curtail their spread to the public. ONT is faster than traditional methods and thus offers a better alternative regarding time, since ONT requires around 6 h compared with the longer time required for the conventional method, which is based on culture, biochemical tests and antibiotic disk susceptibility tests [33]. Moreover, ONT sequencing offers unique advantages over short read-based sequencing methods, including lower equipment costs, real-time base-calling, the enrichment of samples via adaptive sampling and high portability. Likewise, surveillance based on genomics detects more AMR in ONT compared with what is obtainable for the conventional method, which is currently and frequently being used in Nigeria. To date, ONT MinION sequencer has been widely applied to research to detect bacteria, parasites and viruses in both clinical and environmental samples [34,35,36]. The MinION device has been used in detecting bacterial pathogens and AMR genes in clinical isolates [19,37,38].

There are several public health and economic consequences associated with the AMR crisis and the animal health sector is pivotal in its management. Indeed, resistant pathogens can be cross-circulated via animal–environment–human pathways with the possibility to acquire and/or disseminate ARGs through both horizontal and vertical transfer [39,40]. Worse yet, AMR monitoring and stewardship strategies in sub-Saharan Africa are still ineffective, making this part of the world the most affected by this scourge. In this study, we observed a high resistance to ciprofloxacin (70%, n = 7), which is commonly used to treat respiratory, enteric and urinary tract infections but is also used as growth promoter in animal production. Apart from Penicillin G (100%, n = 10), other critical resistances were represented by tetracycline (50%, n = 4), cefazolin (40%, n = 4) and sulfonamide (30%, n = 3). Most of these antibiotic categories are largely used in food production in Nigeria [41]. Focusing this study on *E. coli* is valuable as it is a common contaminating organism in animal-derived food, especially in the LMICs, and is also used to monitor AMR burden and transmission across the different sectors [42,43]. It is a major reservoir of resistance genes, which may be responsible for treatment failures in both human and veterinary medicine [44]. In addition, the prevalence and AMR levels in *E. coli* represent a good indicator of the ‘selective pressure’ among antimicrobials commonly used in food-producing animals either as growth promoters, prophylaxis or treatment [45]. In general, antimicrobial resistance in *E. coli* is considered one of the major worldwide challenges in both animals and humans [44].

To the best of our knowledge, this is the first study on the nanopore-driven genomic analysis of *E. coli* isolates from animals in Nigeria. However, this study’s findings have been impacted due to some methodological limitations. Despite the versatile advantages of long reads, unlike short read-based sequencing, the downstream analysis is not that robust as the nanopore sequencing method is prone to high read-level error rates. One of the major drawbacks of the current study is that short read sequencing was not performed as a comparator towards determining the efficiency of the long read-based analysis of the ARGs in the isolates. Moreover, the assemblies were not robust as the read lengths were below 1000 bp, eventually overlooking plasmid-associated ARGs. In addition, the sequences were not filtered and the assemblies were not polished, which is imperative for accurate analysis. In a recent study, the polishing of de novo assemblies has been found to be indispensable in improving ARGs analysis with long read sequences [46]. Another study reported the higher performance of hybrid assembled genomes in predicting ARGs, plasmids and virulence factors [47]. Owing to the above limitations, the discrepancy between the phenotype and genotype was found to be high. In addition, the plasmids containing ARGs were not detected either, which can be attributed to the short length of the long reads. Apart from the limitations in study design and analysis, the use of ONT and advanced diagnostic kits might be a challenge for the LMIC settings due to delays in reagent procurement, but also the scarcity of qualified staff for sequencing and bioinformatic analysis. Finally, the sample size in the current study was very limited and isolates were randomly chosen for sequencing. The inclusion of more isolates with a systematic selection from both pigs and poultry layers would have increased the robustness of the analysis in making a good correlation between phenotype and genotype, ensuring a generalizable finding. Notwithstanding the mentioned limitation, this sequencing study will be a precursor towards accentuating future studies to explore the AMR genotypes in bacterial population from different sectors.

The findings of this study, again, substantiate the rational use of antibiotics, including ciprofloxacin, tetracycline and cefazolin, which are commonly used as first-line treatments in both the animal and human health sectors in Nigeria. Despite potential limitations, the study supports the application of nanopore sequencing for AMR monitoring in low-resource settings. However, the further optimization of the method with a large sample size is highly appreciated to establish a good correlation between the phenotype and genotype of the bacterial isolates. Recent studies showed the promises of nanopore sequencing in the real-time monitoring of resistance. Further large-scale studies are warranted to develop a robust sequence analysis pipeline and the optimization of the existing analysis tools to streamline the deployment of nanopore sequencing in LMICs for AMR surveillance.

## Figures and Tables

**Figure 1 antibiotics-14-00827-f001:**
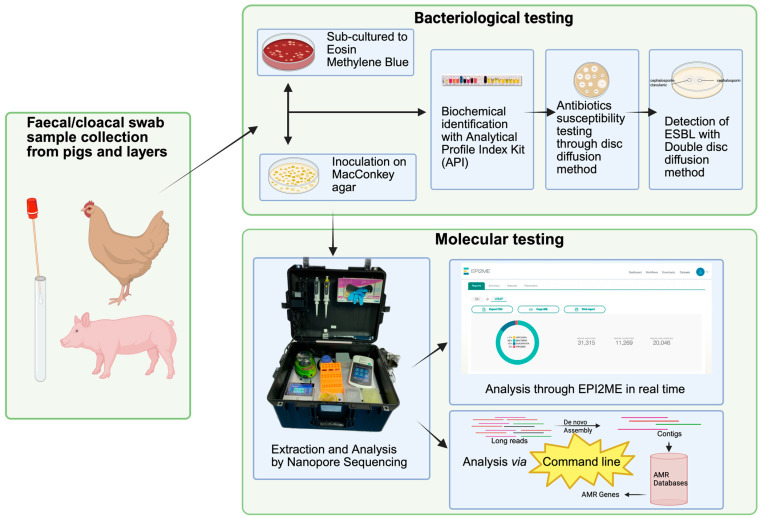
Schematic diagram depicting sample collection, bacteriological testing and nanopore sequencing followed by downstream analysis.

**Figure 2 antibiotics-14-00827-f002:**
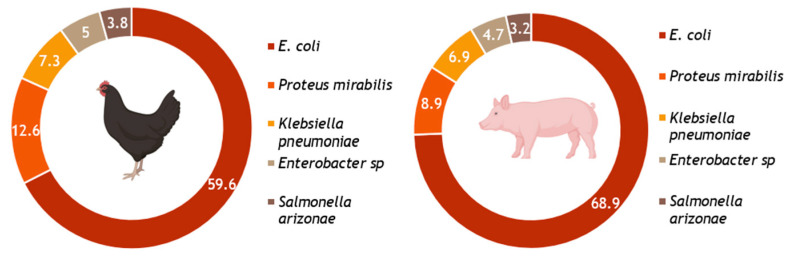
Distribution of the predominant enterobacterial species isolated from pigs and chickens.

**Figure 3 antibiotics-14-00827-f003:**
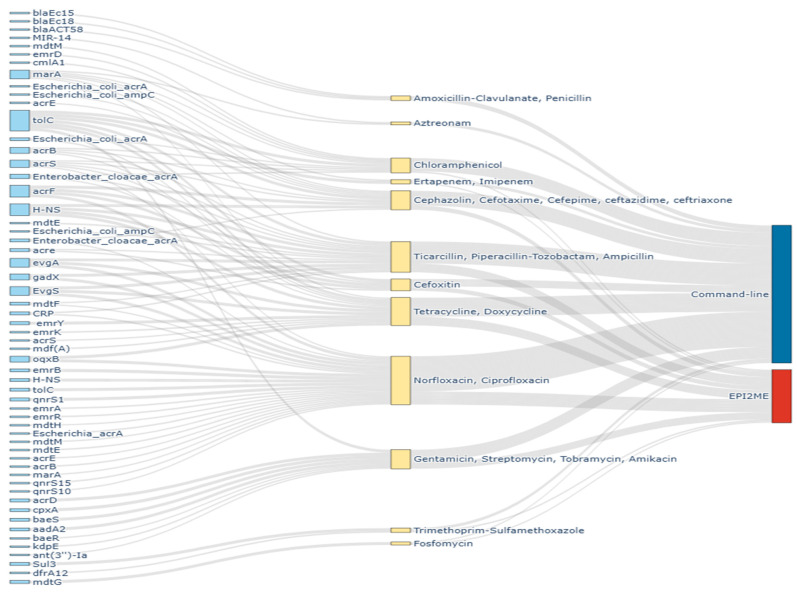
Distribution of the ARGs according to the drug families following the analytical process.

**Figure 4 antibiotics-14-00827-f004:**
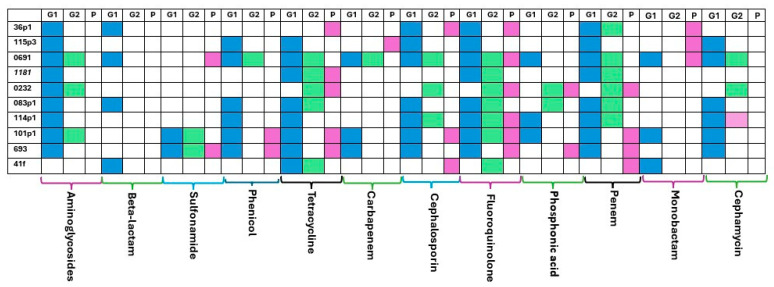
Cross comparison between genotyping (G1 and G2) and phenotyping (P) approaches for the 10 *Escherichia coli* isolates. Blue color represents the presence of ARGs for G1; green color represents the presence of ARGs for G2 and purple color represents the phenotypic resistance against each antibiotic class.

**Table 1 antibiotics-14-00827-t001:** Results of antimicrobial susceptibility test on *Escherichia coli* isolates.

Antibiotic Class	Antibiotics	Concentration (µg/disk)	Number of Resistant *E. coli* (*n* = 10)
Aminoglycoside	Gentamicin (CN)	(10 µg)	0 (0%)
	Amikacin (AK)	(30 µg)	1 (10%)
	Streptomycin (S)	(25 µg)	1 (10%)
	Tobramycin (TOB)	(10 µg)	0 (0%)
Phosphonic acid	Fosfomycin (FOS)	(50 µg)	0 (0%)
Beta lactam/beta lactam Inhibitor	Penicillin (P)	(10 µg)	10 (100%)
	Amoxicillin/Clavulanate (AMC)	(30 µg)	2 (20%)
Penem	Ampicillin (AMP)	(10 µg)	4 (40%)
	Ticarcillin (TIC)	(75 µg)	3 (30%)
	Piperacillin/Tazobactam (TZP)	(110 µg)	0 (0%)
Cephalosporin	Cefotaxime (CTX)	(30 µg)	1 (10%)
	Ceftazidime (CAZ)	(30 µg)	1 (10%)
	Cefepime (FEP)	(30 µg)	1 (10%)
	Cephazolin (KZ)	(30 µg)	4 (40%)
	Ceftriaxone (CRO)	(30 µg)	2 (20%)
Cephamycin	Cefoxitin (FOX)	(30 µg)	0 (0%)
Monobactam	Aztreonam (ATM)	(30 µg)	3 (30%)
Carbapenem	Ertapenem (ETP)	(10 µg)	1 (10%)
	Imipenem (IPM)	(10 µg)	0 (0%)
Fluoroquinolone	Ciprofloxacin (CIP)	(5 µg)	7 (70%)
	Norfloxacin (NOR)	(10 µg)	0 (0%)
Phenicol	Chloramphenicol (C)	(30 µg)	2 (20%)
Sulfonamide	Sulfamethoxazole/Trimethoprim (SXT)	(25 µg)	3 (30%)
Tetracycline	Tetracycline (TE)	(30 µg)	5 (50%)
	Doxycycline (DO)	(30 µg)	5 (50%)

## Data Availability

The sequencing datasets generated and analyzed during the current study are available in the Sequence Read Archive (SRA) with the submission number, SUB14898118, and the following web link: https://submit.ncbi.nlm.nih.gov/subs/sra/SUB14898118/overview, accessed on 10 May 2025. The accessions are SAMN45815315, SAMN45815316, SAMN45815317, SAMN45815318, SAMN45815319, SAMN45815320, SAMN45815321, SAMN45815322, SAMN45815323 and SAMN45815324.

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
