# Peer review of "Nanopore Sequencing-Driven Mapping of Antimicrobial Resistance Genes in Selected Escherichia coli Isolates from Pigs and Poultry Layers in Nigeria"

_antibiotics, 2025, doi:10.3390/antibiotics14080827_

Round 1
Reviewer 1 Report
Comments and Suggestions for Authors
Nanopore sequencing driven mapping of the antimicrobial resistance genes in Escherichia coli from pigs and poultry layers in Nigeria
Specific Comments
Line 98-103: “Interestingly, it is possible to rapidly identify pathogens in 10 minutes after sequenc- ing and the detection of all pre-defined AMR encoding genes and plasmids (17). The same feat could be achieved from a non-culture experiment within 1 hour by using nanopore sequencing data (17). The short turnaround time for ONT, compared to a period of no less than 4 days required for conventional cultural identification and sensitivity methods, demonstrates the advantage of the former over the latter.”.
Comment: “Can you paraphrase this paragraph for clarity?”.
Line 188: “Data processing with command line tools.”.
Comment: “More detail explanation on command line tools is needed.”.
Suggestions: “Content for this section seems missing”
Line 223: “A filter was used to highlight clinically relevant results”.
Comment: “How can you describe clinically relevant results”.
Figure 3 seems bright. Can you use light colored palettes.
General Comments
- Clarity: The writing and flow of the manuscript seems clear to me.
- Methodology: Methodology was well described with schematic diagram.
- Results: Results were well presented.
- Discussion: Elaborate more on limitations of the study.
Author Response
Reviewer 1
Specific Comments
Line 98-103: “Interestingly, it is possible to rapidly identify pathogens in 10 minutes after sequencing and the detection of all pre-defined AMR encoding genes and plasmids (17). The same feat could be achieved from a non-culture experiment within 1 hour by using nanopore sequencing data (17). The short turnaround time for ONT, compared to a period of no less than 4 days required for conventional cultural identification and sensitivity methods, demonstrates the advantage of the former over the latter.”
Response: Thanks a lot for your remarks towards improving the manuscript.
Comment: “Can you paraphrase this paragraph for clarity?”
Response: Thanks for your comment. The section has been revised.
Line 188: “Data processing with command line tools.”
Comment: “More detailed explanation on command line tools is needed.”.
Suggestions: “Content for this section seems missing”
Response: Thanks for the comment. It was revised.
Line 223: “A filter was used to highlight clinically relevant results”.
Comment: “How can you describe clinically relevant results”.
Response: The clinically relevant results have been described in the context of observable phenotypic resistance to antibiotics that are routinely used for treatments of infections in animals and humans in Nigeria, including antibiotics from important groups such as fluoroquinolones and cephalosporins.
Figure 3 seems bright. Can you use light colored palettes?
Response: Figure revised.
General Comments
- Clarity: The writing and flow of the manuscript seems clear to me.
- Methodology: Methodology was well described with a schematic diagram.
- Results: Results were well presented.
- Discussion: Elaborate more on limitations of the study.
Reviewer 2 Report
Comments and Suggestions for Authors
This manuscript describes the isolation and genome sequencing of E. coli from pigs and poultry layers in Nigeria. Nanopore genome sequencing is performed on 10 isolates, and the authors aim to explore the antibiotic resistance gene portfolio.
While it’s crucial to study bacterial pathogens and their carriage of antibiotic resistance genes, particularly in countries like Nigeria, the manuscript has several critical issues that might not be resolved (or require strong clarification). There are conflicts in the methods and results described. Additionally, the data should be presented in a logical manner. Several typos could be identified.
Major comments:
- While the long-read Nanopore sequencing has been carried out, why do you get reads with an average length of ~893.5 and 1482 bp? This is very strange.
- When the swab samples were taken and bacteria ( coli) were isolated, how did you make sure that there was only a single type of E. coli? Rectal swabs should give you different types of E. coli strains. A random selection of E. coli and its genome sequencing would not represent a comprehensive scenario. Also, was the antibiotic resistance and sequencing carried out using the same E. coli colony or strain?
- How many total bacteria were isolated? Why and using which criteria did you choose 10 of those many isolates?
- The genome sequencing statistic is completely missing. How many reads were generated per sample? How many reads make up an assembly? How many contigs? How many of these contigs were of plasmids?
- Were there any plasmids? Plasmids is an essential factor in relation with Abr genes. This should have been described.
- The results could have been presented better. For e.g. Table 2 should be completely replaced by a plot.
- “…Command line and EPI2ME…” is not the standard practice to describe the assembly methods. These should be linked to tool and version for e.g., Flye v2.1 and EPI2ME pipeline version.
- Sequenced bacteria should be re-identified based on the genome, as phenotypic data is not always accurate.
Other minor (listing a few of many)
- In the abstract, 94 ARGs were mentioned, and in the results, 95 ARGs (L262, L275).
- In the methods, a new database is created for the Abr genes, but the results described “two pipelines used were able to predict ARGs…. resistance of the identified genes being retrieved from the CARD database…”??
- Too much epidemiological introduction for the Nanopore genome sequencing-based analysis, should be reduced to a maximum to one paragraph and more content is needed about the genome sequencing data available from Nigeria.
- “Phred-quality score should be ≥7”, why “should be”? Did you confirm?
- Conflicting “…..read size ≥1000 nucleotides. Since most of reads were less than 1000 nucleotides, all were considered for analysis, regardless of their size.” So many reads were less than 1000 and considered for the study? You also mentioned ~893.5 average basepair read length in the manuscript.
- “2 - 4 pure isolated colonies were picked” for genome sequencing? Are these colonies coming from stock or direct swabs?
Author Response
Reviewer 2
Comments and Suggestions for Authors
This manuscript describes the isolation and genome sequencing of E. coli from pigs and poultry layers in Nigeria. Nanopore genome sequencing is performed on 10 isolates, and the authors aim to explore the antibiotic resistance gene portfolio.
While it’s crucial to study bacterial pathogens and their carriage of antibiotic resistance genes, particularly in countries like Nigeria, the manuscript has several critical issues that might not be resolved (or require strong clarification). There are conflicts in the methods and results described. Additionally, the data should be presented in a logical manner. Several typos could be identified.
Response: Thanks a lot for your remarks towards improving the manuscript.
Major comments:
- While the long-read Nanopore sequencing has been carried out, why do you get reads with an average length of ~893.5 and 1482 bp? This is very strange.
Response: Yes, you are right. The purpose of nanopore sequencing is to get long reads over the bacterial genome. However, it is not unlikely to get poor quality or small reads. In our study we found poor quality reads which can be attributed to poor quality of genomic DNA, suboptimal library preparation and poor efficiency of the flow cell while performing the sequencing. Given the poor quality of the sequencing data, we considered all of the reads below 1000bp for downstream analysis to capture optimum information towards genotyping of the bacterial isolates and detection of the ARGs in them. To enhance the robustness of the analysis, we applied the proper sequence filtering algorithm and removed the unwanted reads.
- When the swab samples were taken and bacteria ( E. coli) were isolated, how did you make sure that there was only a single type of E. coli? Rectal swabs should give you different types of E. coli strains. A random selection of E. coli and its genome sequencing would not represent a comprehensive scenario. Also, was the antibiotic resistance and sequencing carried out using the same E. coli colony or strain?
Response: These has been addressed in the manuscript, standard bacteriological procedures were followed for the identification of enterobacterials and appropriate selective media and API biochemical methods was used to identify Escherichia coli, based on similar morphologic and antibiotic sensitivity result, discrete colonies of the E coli were stored for molecular characterisation.
- How many total bacteria were isolated? Why and using which criteria did you choose 10 of those many isolates?
Response: The various bacteria species from the family enterobacteriacea that were isolated had been incorporated into the manuscript. E coli was the predominant species isolated both from the chickens and pig. Based on the phenotypic resistance pattern to antibiotics of interest (drug of last resort in clinical practice in Nigeria both in animal and humans, such as fluoroquinolones and cephalosporins), ten were selected randomly for molecular analysis.
- The genome sequencing statistic is completely missing. How many reads were generated per sample? How many reads make up an assembly? How many contigs? How many of these contigs were of plasmids?
Response: The detailed statistics on the reads, contigs and plasmids is given in the supplementary table.
- Were there any plasmids? Plasmids are an essential factor in relation with Abr genes. This should have been described.
Response: Replicons were retrieved from 4/10 isolates, including colpVC_1, Col(MG828)_1, Col156_1 and IncN_1, which may play a critical role on ARGs exchange. IncN_1, a conjugative plasmid, is a major concern due to its potential implication on inter-species transmission. Although often non-conjugative, Col replicons may coexist with resistance-bearing plasmids and contribute to their stability in the bacteria.
- The results could have been presented better. For e.g. Table 2 should be completely replaced by a plot.
Response: The table is replaced by a heatmap figure.
- “…Command line and EPI2ME…” is not the standard practice to describe the assembly methods. These should be linked to tool and version for e.g., Flye v2.1 and EPI2ME pipeline version.
Response: You are right. In command line Flye v2.1 was used to make the long read based assembly of the bacterial genome whereas the EPI2ME platform matched the reads with the ARGs sequences from the CARD database. Therefore, the two methods were different in terms of sequence analysis. In EPI2ME, the WIMP (What is in my pot) pipeline was used which matched the reads with the ARGs sequences in the CARD database. The version of the Flye is mentioned in the manuscript.
- Sequenced bacteria should be re-identified based on the genome, as phenotypic data is not always accurate.
Response: Thanks for the remarks. More details are provided on the species identification method.
Other minor (listing a few of many)
- In the abstract, 94 ARGs were mentioned, and in the results, 95 ARGs (L262, L275).- Response: Revision has been made.
- In the methods, a new database is created for the Abr genes, but the results described “two pipelines used were able to predict ARGs…. resistance of the identified genes being retrieved from the CARD database…”??
Response: Both command line and EPI2ME were used to predict the ARGs. Later on, the detected ARGs were classified according to the gene ontology of the CARD database.
- Too much epidemiological introduction for the Nanopore genome sequencing-based analysis, should be reduced to a maximum to one paragraph and more content is needed about the genome sequencing data available from Nigeria.
Response: There are more epidemiological data, however information on sequence types and resistance gene data from animals in Nigeria has been included in the manuscript as requested by the reviewer.
- “Phred-quality score should be ≥7”, why “should be”? Did you confirm?(specific Phred score). Phred-quality score was ≥7”,
Response: Default setting was applied for QC checking and filtering with Phred score ≥7. Text has been rewritten for more understanding.
- Conflicting “…..read size ≥1000 nucleotides. Since most of the reads were less than 1000 nucleotides, all were considered for analysis, regardless of their size.” So many reads were less than 1000 and considered for the study? You also mentioned ~893.5 average base pair read length in the manuscript.
Response: Already described earlier against the major comment-1.
- “2 - 4 pure isolated colonies were picked” for genome sequencing? Are these colonies coming from stock or direct swabs?
Response: This has been addressed in the manuscript. The same bacteria strains that were identified and tested for the antibiotic sensitivity were used for the genomic studies.
Reviewer 3 Report
Comments and Suggestions for Authors
Dear authors,
The study addresses a critical public health issue: antimicrobial resistance (AMR) in livestock in Nigeria. The application of Nanopore sequencing for rapid AMR monitoring is innovative and suitable for resource-limited settings.
The usefulness of Nanopore for LMICs is highlighted due to its portability and speed.
The experimental design is well described, with clear ethical protocols and standard methodology for isolation, AST (susceptibility testing), and sequencing.
Comments:
- Was the assemblies polished?
-Could short reads (<1000 bp) explain the lack of plasmid detection? This should be discussed since genes associated with MGEs are mentioned.
- The implications of the lack of ESBLs should be discussed.
- The statement: AMR prediction was accurate in 16 cases (57.14%) with the command-line approach and in 9 cases (32.14%) using EPI2ME should be discussed. Are the detected genes pseudogenes? Are there flaws in the EPI2ME databases?
- It would be necessary to delve deeper into how these findings can guide antibiotic use policies in Nigeria.
Finally, the manuscript has English and style errors, some of which are:
Abstract: "maneuvering the advanced technologies." Suggestion: "leveraging advanced technologies."
Line 29: "reported previously as transferred through MGEs." Suggestion: "previously reported to be transmitted via MGEs."
Line 35: "has the promises for." Should be: "holds promise for"?
"E. coli" should be written as "E. coli" (italic).
Resistance genes in lowercase and italics
The article is interesting, but requires minor revisions and a more critical discussion of its limitations. I consider it acceptable with modifications.
Comments on the Quality of English LanguageRequires minor grammar and style corrections for greater fluency.
Author Response
Reviewer 3
Comments and Suggestions for Authors
Dear authors,
The study addresses a critical public health issue: antimicrobial resistance (AMR) in livestock in Nigeria. The application of Nanopore sequencing for rapid AMR monitoring is innovative and suitable for resource-limited settings.
The usefulness of Nanopore for LMICs is highlighted due to its portability and speed.
The experimental design is well described, with clear ethical protocols and standard methodology for isolation, AST (susceptibility testing), and sequencing.
Response: Thanks a lot for your remarks towards improving the manuscript.
Comments:
- Was the assemblies polished?
Response: We avoided polishing the assembly because the read quality was poor. The rigorous polishing steps reduce the length of the contig, thus the chance of losing the ARGs would be high.
- Could short reads (<1000 bp) explain the lack of plasmid detection? This should be discussed since genes associated with MGEs are mentioned.
Response: You are right. Due to the poor quality of the reads, we were unable to detect plasmids in all of the isolates. We found a few replicons of plasmids. However, we could not characterise the plasmids and locate the ARGs in them.
- The implications of the lack of ESBLs should be discussed.
Response: The sample size in the current study was too small to capture the exact prevalence of ESBL E coli in the animal sector in Nigeria. Given that limitation, to get more generalised epidemiological information on the prevalence of ESBL E coli further large scale studies are warranted.
- The statement: AMR prediction was accurate in 16 cases (57.14%) with the command-line approach and in 9 cases (32.14%) using EPI2ME should be discussed. Are the detected genes pseudogenes? Are there flaws in the EPI2ME databases?
Response: In command line the whole genome assembly is developed from the reads and then AMR genes are detected through comparing with the known database. On the other hand, in the EPI2ME WIMP pipeline, reads are aligned with the AMR gene database/CARD database. In command line multiple databases were used to retrieve the AMR genes whereas only CARD was used in EPI2ME to detect AMR genes. Eventually, more ARGs were detected in command line. However, a protein homolog model with >90% accuracy was used for EPI2ME, therefore the chances of detecting pseudogenes has been low.
- It would be necessary to delve deeper into how these findings can guide antibiotic use policies in Nigeria.
Response: In this study, only 10 E. coli isolates were used to make a correlation between the genotypes and phenotypes. Therefore, it is too early to infer from the genotype results which can be translated into policy. However, the findings of the phenotype and the implication on the use of antibiotics in both human and animal sectors have been discussed already.
Finally, the manuscript has English and style errors, some of which are:
Abstract: "maneuvering the advanced technologies." Suggestion: "leveraging advanced technologies."
Response: Corrected.
Line 29: "reported previously as transferred through MGEs." Suggestion: "previously reported to be transmitted via MGEs."
Response: Corrected.
Line 35: "has the promises for." Should be: "holds promise for"?
Response: Corrected.
"E. coli" should be written as "E. coli" (italic).
Response: Corrected.
Resistance genes in lowercase and italics
The article is interesting, but requires minor revisions and a more critical discussion of its limitations. I consider it acceptable with modifications.